# SAGE CAN QUANTIFY WHY TWO MODELS BEHAVE DIFFERENTLY

## ABSTRACT

Vision-based activity recognition tasks are sensitive to environmental context and lighting, making generalization across domains difficult. Models trained in controlled settings can report high accuracy, but often fail under domain shift, where it remains unclear whether predictions depend on causal foreground cues, spurious background signals, or shortcut learning tied to context rather than behavior. Saliency methods offer a view of model focus, but have largely been confined to qualitative visualization. We hypothesize that behavioral divergence between models is proportional to divergence in their saliency embeddings. To examine this, we introduce Saliency Attribution for Goal-grounded Evaluation (SAGE), a modular framework that unifies heterogeneous datasets through category mapping and balancing, generates controlled foreground and background variants, computes saliency maps, and encodes them into tokenized representations suitable for embedding and comparison. By disentangling foreground and background saliency, the framework provides a diagnostic signal of how models attend to causal versus spurious regions, complementing accuracy as a measure of generalization. We demonstrate feasibility on vision-based driver distraction detection, an activity recognition task where distraction is inferred from driver activities rather than objects, by creating a unified 10-class variant of the StateFarm and 100-Driver datasets that highlights the challenges of category mapping and background control. While full embedding-based evaluations are ongoing, the framework separates foreground and background saliency, discretizes them into tokens, and encodes them in a manner aligned with tokenized vision architectures such as ViTs and VLMs. This design makes the framework scalable across vision-based classification tasks where foreground-background disentanglement is critical, and presents it as a diagnostic tool for analyzing behavioral divergence and robustness under domain shift.

**Keywords:** Explainable AI, Vision-based Driver Distraction Detection (vDDD), SAGE, Saliency Embeddings, Behavioral Divergence, Domain Shift, Generalization, Shortcut Learning, Vision–Language Models (VLMs)

## 1 INTRODUCTION

Activity recognition in vision systems remains fragile under domain shift, where models trained in controlled environments fail when deployed in settings with different lighting, context, or subject distributions (Duan et al., 2023). A key uncertainty is whether predictions depend on causal action cues (e.g., hand and torso dynamics) or spurious background signals (e.g., dashboard texture, seat position) (Geirhos et al., 2020). Conventional saliency methods such as Grad-CAM (Selvaraju et al., 2017) provide qualitative insights into model focus but lack a standardized representation for comparison, limiting their use in diagnosing or quantifying behavioral divergence.

We introduce **Saliency Attribution for Goal-grounded Evaluation (SAGE)**, a modular framework that moves saliency from visualization to structured diagnostics. SAGE unifies heterogeneous activity datasets through category mapping (Montoya et al., 2016; Wang et al., 2023), generates controlled background variants (full, bbox, segmentation, bbox-guided segmentation), computes normalized saliency maps across multiple CAM methods, and converts them into foreground- and background-aware token descriptors. These tokens capture patch-level statistics (mass, centroid, maxima, foreground fraction) over grids (e.g., $4{\times}4$, $8{\times}8$), enabling embeddings that align naturally with patch-based architectures and multimodal encoders (Dosovitskiy et al., 2021).

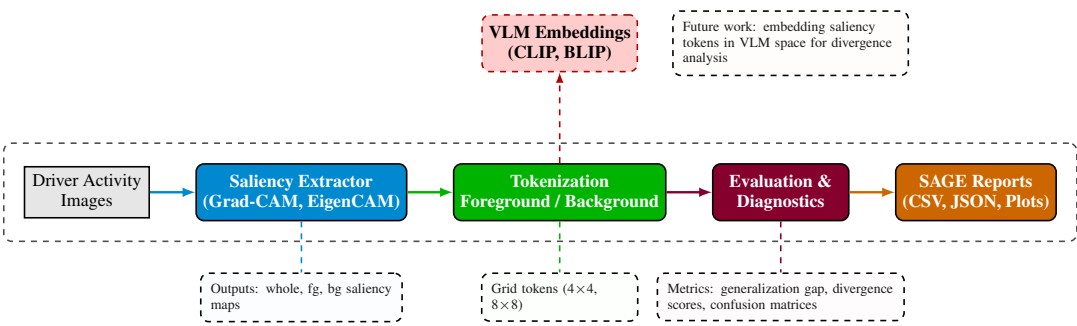

Figure 1: Overview of the **SAGE** framework. The dashed box marks the *current scope* (SAGE v1.0): saliency extraction, tokenization, and evaluation pipelines producing quantitative reports. The VLM embeddings module is out of scope in current work and represents future work.

Our experimental study uses driver distraction detection as a representative domain, harmonizing the StateFarm dataset with the 100-Driver dataset into a ten-class schema. We benchmark EfficientNet-B0 and lightweight CNNs across cross-domain settings, with models trained on one dataset variant and evaluated on another. Results show asymmetric degradation: symmetric gestures (`talking left/right`) and occlusion-heavy actions (`reaching behind`, `makeup`) collapse disproportionately, while distinct actions (`texting`, `safe driving`) transfer more reliably. Saliency mosaics and token statistics confirm that correct predictions align with foreground cues, while errors diffuse into background regions. Variants that suppress background (`SEG`, `BBOXSEG`) reduce shortcut reliance but do not eliminate failures, highlighting the persistent role of symmetry, occlusion, and rare-class drift.

By discretizing saliency into reproducible token representations, SAGE complements accuracy with a diagnostic lens: divergence between models can be attributed to whether they anchor on causal or spurious regions. This design not only advances driver distraction detection but also provides a generalizable template for activity recognition tasks where foreground-background disentanglement is critical (Moayeri et al., 2022; Xiao et al., 2020; Fang et al., 2018). Figure 1 illustrates the workflow spanning dataset unification, saliency computation, tokenization, and downstream analysis.

## 2 RELATED WORK

**Cross-dataset generalization.** Cross-dataset generalization has been recognized as a critical requirement for deploying vision-based driver distraction detection (vDDD) in real-world conditions. Models trained on one dataset often degrade significantly when evaluated on another due to distributional shift. Duan et al. (2023) introduced a Score-Softmax classifier to enhance cross-dataset performance in distracted driving detection, and Zandamela et al. (2022) emphasized out-of-distribution (OOD) testing as a benchmark of robustness. While such works improve accuracy, they mainly report *what* changes across domains rather than *why*, leaving open the question of which features drive divergence.

**Foreground-background sensitivity.** Background has been argued to act as either noise or useful context depending on the setting. Xiao et al. (2020) showed that removing background can sometimes improve robustness, while Moayeri et al. (2022) demonstrated that discarding it indiscriminately can harm performance, since not all background is spurious. These findings motivate approaches that can disentangle causal contributions from foreground versus background rather than treating the latter uniformly as noise.

**Causal feature alignment.** To address this challenge, Venkataramani et al. (2024) proposed Causal Feature Alignment (CFA), explicitly aligning model representations with causal regions. CFA cautions that indiscriminate background removal can degrade performance when background carries causal information. However, while alignment is promising, systematic diagnostics to reveal when models rely on causal versus spurious features across domains remain limited.

**Saliency as explanation.** Saliency-based explanations are widely used to analyze where models focus. Early methods such as gradient saliency (Simonyan et al., 2014) laid the foundation, followed by Grad-CAM (Selvaraju et al., 2017) and Grad-CAM++ (Chattopadhay et al., 2018), which became de facto visualization tools in vDDD and beyond. These methods show whether robust models attend

more to the driver foreground rather than background, but outputs are qualitative, layer-sensitive, and difficult to compare across datasets.

**Trustworthiness of saliency.** The reliability of saliency methods has been repeatedly questioned. Adebayo et al. (2018) revealed that many attribution techniques fail sanity checks, while Kindermans et al. (2017) showed their sensitivity to small perturbations. More recently, Venkatesh et al. (2024) argued that gradient-based saliency maps cannot be trusted as faithful indicators of model reasoning. These critiques underscore the limitations of saliency as an explanatory tool, especially in robustness evaluation.

**Saliency in training.** Beyond post-hoc visualization, some works attempt to integrate saliency into training pipelines. Van Zyl et al. (2024) and Tempel et al. (2025) proposed attribution-guided regularizers to steer models toward human-interpretable features, while Dey et al. (2021) investigated context-driven distraction detection with saliency emphasis. Despite these attempts, the effectiveness of saliency-guided training remains inconsistent without causal grounding.

**From human fixation to model saliency.** Human saliency prediction has established standardized datasets and metrics to benchmark computational models of visual attention (Judd et al., 2012; Borji et al., 2013). These works define a rigorous pipeline, saliency as a concept, represented by maps, evaluated via metrics, that has enabled reproducible comparisons. Model saliency, however, differs fundamentally: it reflects neural network focus rather than human gaze. Borrowing the benchmarking mindset while adapting it to model saliency remains unresolved.

**Architectural trends.** Advances in architectures extend saliency analysis beyond CNNs. Vision Transformers and vision-language models require new attribution techniques, with Zheng et al. (2025) examining saliency in VLMs for safety-critical applications. These developments show that semantic embeddings can enrich explanatory signals, but their integration into cross-dataset vDDD analysis is still lacking.

## 3 FRAMEWORK

We introduce a modular pipeline for activity recognition that converts saliency maps into *foreground/background-aware token vectors* and standardizes downstream diagnostics under a declarative, reproducible workflow. The key contribution is the formulation and implementation of saliency tokenization and analysis at scale.

**Problem conceptualization:** Let $\mathcal{D} = \{(x_i, y_i)\}_{i=1}^N$ with images $x_i \in \mathbb{R}^{H \times W \times 3}$ and labels $y_i \in \mathcal{Y}$. A classifier $f_\theta$ produces logits $z = f_\theta(x)$ and $\hat{y} = \arg\max_k z_k$. A saliency method $\ell$ yields a map $S(x; f_\theta, \ell) \in \mathbb{R}^{H \times W}$. Foreground/background masks $M_{\text{fg}}, M_{\text{bg}}$ are defined as in Sec. **Dataset preparation**. We rectify and normalize saliency via

$$\tilde{S} = \frac{\phi_{\text{ReLU}}(S)}{\sum_{u,v} \phi_{\text{ReLU}}(S_{u,v})}, \quad \phi_{\text{ReLU}}(t) = \max(0, t),$$

yielding a spatial probability distribution with $\sum \tilde{S} = 1$. Foreground and background saliency masses are $A_{\text{fg}} = \sum \tilde{S} \odot M_{\text{fg}}$ and $A_{\text{bg}} = \sum \tilde{S} \odot M_{\text{bg}}$. A patch grid $\mathcal{G}$ (e.g., $P \times Q$) defines per-cell descriptors

$$\mathbf{t}_g = \Big[ \sum_{(u,v) \in g} \tilde{S}_{u,v}, \ \text{mean}_g(\tilde{S}), \ \max_g(\tilde{S}), \ \text{COM}_g(\tilde{S}), \ \text{fgfrac}_g \Big],$$

where $\text{fgfrac}_g = \frac{\sum_{(u,v) \in g} M_{\text{fg}}}{|g|}$. The set $\{\mathbf{t}_g\}_{g \in \mathcal{G}}$ constitutes the *saliency tokenization*, stored verbatim for downstream embedding.

**Dataset preparation:** Cross-domain evaluation requires a unified label space. We harmonize the State Farm Distracted Driver dataset (SF3D) (Montoya et al., 2016) and the 100-Driver dataset (Wang et al., 2023) into a ten-class schema via a declarative mapping (Appendix 2). Many-to-one merges preserve class balance through explicit weights. Foreground masks are acquired via pluggable detectors: YOLOv8 (COCO-pretrained `person`) in this work, but extendable to SAM2 or VLM-based selectors. Four anchored variants are generated: FULL (unaltered frame), BBOX (driver crop), SEG (segmented foreground), and BBOXSEG (bbox-guided segmentation). Each variant preserves original splits and produces aligned artifacts: cropped/masked RGBs, binary masks, YOLO+VIA annotations, and per-split statistics. Figure 2 illustrates mosaics for SF3D and the unified 100-driver-sf3d-nc10.

**Training and evaluation:** A declarative loader feeds a common training/evaluation stack across 26 torchvision CNNs. We benchmark five lightweight backbones DenseNet-121, EfficientNet-

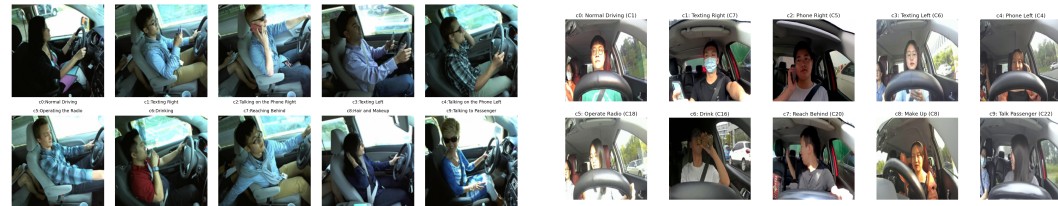

<p style="text-align:center">(a) SF3D mosaic          (b) 100-Driver remapped (nc10)</p>

Figure 2: Mosaics of the ten-class activity label space. Panel (a) shows the SF3D categories, while panel (b) shows the corresponding remapped classes from the 100-Driver dataset. The side-by-side view highlights semantic alignment and cross-dataset variability.

B0, ResNet-18, ResNet-50, and SqueezeNet-1.0 ($224 \times 224$) motivated by cross-domain generalization studies (Wang et al., 2023; Duan et al., 2023), while an additional 21 models (e.g., DenseNet-161/169/201, Inception-v3, MobileNet-v2/v3, VGG-11/13/16/19, ShuffleNet-v2, ResNet-34/101/152, Wide-ResNet-50/101) can be enabled without code changes. All models use ImageNet-pretrained weights, cross-entropy loss, batch size 32, and 100 epochs. Each run records configuration files, seeds, TensorBoard logs, ROC curves (overall and per-class), confusion matrices with FP/FN image lists, checkpoints (best and final), speed/performance metrics, and a timestamped run directory encoding dataset and variant identifiers. Each dataset variant pair is thus treated as a first-class dataset in the configuration, ensuring systematic coverage and reproducibility.

**Cross-domain inference and evalaution:** Models trained on source variants ($\mathcal{D}^{(a)} \to \mathcal{Y}_U$) are evaluated on target variants ($\mathcal{D}^{(b)} \to \mathcal{Y}_U$), with predictions, confidences, and metrics exported as CSV/JSON for downstream analysis. In practice, we consider three source variants from SF3D-bbox (reduced background), seg (background removed), and bboxseg (background controlled), and evaluate each trained model against multiple target variants from 100-Driver, namely bbox, seg, bboxseg, and full. This design creates a structured $3 \times 4$ evaluation matrix in which each cell corresponds to a distinct pairing of background treatment at training and testing. By traversing this grid, it becomes possible to directly assess the consequences of background mismatches: when a model trained with limited or no background is tested in a background-rich target, or conversely when a background-reliant source model is evaluated on background-suppressed targets.

The purpose of this construction is not to exhaustively enumerate experiments but to provide a principled lens on the role of background in generalization. Consistent structure across variants enables attribution of performance changes to differences in background exposure rather than other confounding factors. Evaluations capture both predictive performance (accuracy, F1, calibration) and representational focus (saliency distributions over foreground and background regions), but the essential methodological point is that the cross-domain grid enforces a controlled comparison. Each row reflects a fixed training condition, while each column modulates the target's background availability. This arrangement provides a compact yet expressive framework for diagnosing whether a model's decision boundary is anchored in causal foreground cues or entangled with spurious background context.

**Confidence-stratified clustering:** To ensure fair saliency comparison, we stratify images by correctness and confidence. For each class $c$, correct and error sets are partitioned into four bins via $k$-means on confidence scores. Clusters are stored with thumbnails and dashboards; saliency is generated within clusters to align difficulty levels.

**Saliency generation and analysis:** We integrate $\approx 14$ methods (Grad-CAM, Grad-CAM++, EigenCAM, ScoreCAM, etc.) via `python-grad-cam`. Outputs are standardized by per-image min–max scaling, ReLU+PDF normalization, and fg/bg masking to compute $A_{\mathrm{fg}}, A_{\mathrm{bg}}$ and patch-level statistics. For each image/method we emit arrays (`whole`, `fg`, `bg`, `mask`, `tokens`) plus overlay PNGs. Token-level analysis computes fg/bg mass ratios, patch-wise KL between variants, cluster-conditioned saliency stability, and rank correlations. Tokens are stored as CSV and `.npy` for efficient reuse.

**Reproducibility:** All stages are declaratively configured (datasets, mappings, saliency methods, grids, layers) via YAML and CLI. Naming follows `logs/{stage}-{timestamp}/{model}-nc{num_classes}-{dataset}-{variant}/`, with model fingerprints, code commits, and seeds recorded. Artifacts are joinable via stable indices (`image_id`, `split`, `class`, `variant`, `method`, `grid`).

Table 1: Aggregated performance metrics of EfficientNet-B0 across dataset variants. Reported metrics: Accuracy ('a'), F1-score ('f1'), Precision ('p'), Recall ('r') in percentages. Each row indicates the dataset the model was trained on and the dataset it was evaluated on. Bold red values denote global bests; underlined values denote per-block bests.

| Variant (Train → Eval) | a | f1 | p | r | Support |
|---|---|---|---|---|---|
| BBoxSeg (SF3D → 100-Driver) | 20.91 | **13.41** | 18.71 | 20.91 | 5255 |
| BBox (SF3D → 100-Driver) | 20.33 | 12.94 | 18.06 | 20.33 | 5255 |
| Seg (SF3D → 100-Driver) | **20.93** | 13.34 | 18.56 | **20.93** | 5255 |
| Full (SF3D → 100-Driver) | 16.69 | 9.40 | 8.98 | 16.69 | 2834 |
| BBoxSeg (100-Driver → SF3D) | 12.91 | **9.64** | 14.12 | 12.91 | 1843 |
| BBox (100-Driver → SF3D) | 12.63 | 9.41 | 14.01 | 12.63 | 1843 |
| Seg (100-Driver → SF3D) | 12.79 | 9.54 | 14.02 | 12.79 | 1843 |
| Full (100-Driver → SF3D) | 11.23 | 8.12 | 13.41 | 11.23 | 1843 |

(a) `efficientnet_b0` trained on SF3D-bboxseg, tested on 100-Driver.

(b) `efficientnet_b0` trained on SF3D-seg, tested on 100-Driver.

Figure 3: Per-class confusion matrices with FP/FN overlays for EfficientNet-B0 variants trained on SF3D and evaluated on 100-Driver (Cam2). Clear misclassification trends appear in symmetric and rare classes.

**Limitations:** This work does not include token embeddings or benchmark scores. However, the design explicitly enables both: $\{\mathbf{t}_g\}$ are embeddable via visual or text prompts, and evaluators already log the metadata needed for goal-conditioned alignment. Current results emphasize robust visualization, normalized tokenization, and reproducible analysis across datasets and variants.

## 4 RESULTS

Class-wise generalization trends were further examined using confusion matrices with false positives (FP) and false negatives (FN) highlighted. Representative EfficientNet-B0 models trained on SF3D and evaluated on 100-Driver (and vice versa) are selected from the `bbox`, `seg`, and `bboxseg` variants. Figures 3a–4b visualize the confusion matrices, revealing asymmetric misclassification patterns and systematic class collapses under domain shift. Frequent activities with strong visual cues (e.g., 'safe driving', 'texting right') show partial transferability, while classes with symmetric gestures (e.g., 'talking left/right') or occlusion-heavy actions (e.g., 'reaching behind') exhibit severe degradation.

The confusion matrices in Figures 3 and 4 demonstrate that domain shift does not uniformly affect all classes. For instance, 'texting right' retains recall above 0.7 across settings, while 'talking left' often collapses with recall below 0.1. Rare categories such as 'makeup' and 'reaching behind' exhibit both low recall and high false positives, especially when evaluated cross-domain. These patterns indicate that model transfer is constrained not only by domain differences but also by intra-class ambiguity and dataset imbalance.

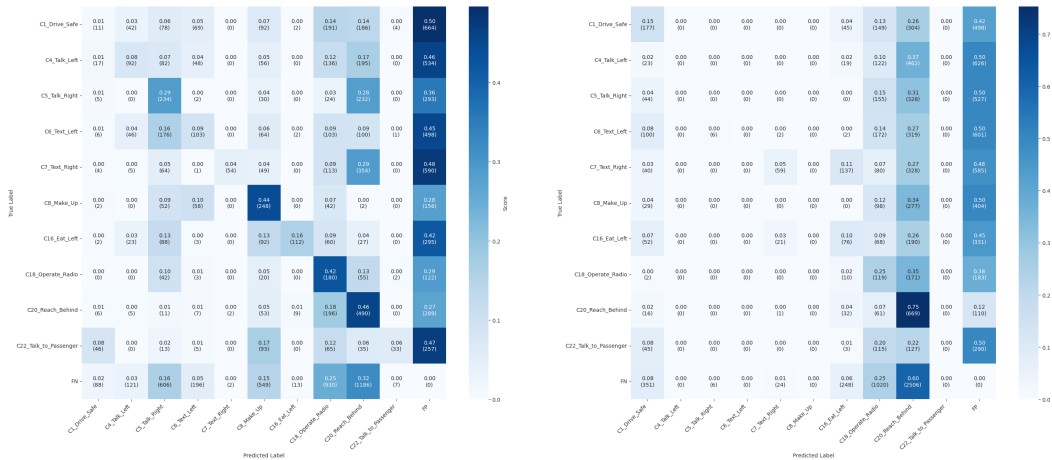

(a) `efficientnet_b0` trained on 100-Driver-bbox, tested on SF3D.

(b) `efficientnet_b0` trained on 100-Driver-seg, tested on SF3D.

Figure 4: Per-class confusion matrices with FP/FN overlays for EfficientNet-B0 variants trained on 100-Driver and evaluated on SF3D. Misclassifications reveal strong dataset-specific biases.

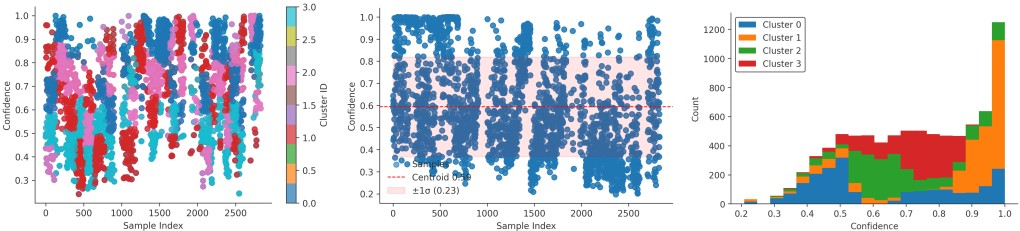

Figure 5: Representative $k$-means ($k{=}4$) confidence-stratified clusters, visualized as scatter, band, and histogram plots.

---

**Key Takeaways on Per-Class Errors and Background Role**

- **Error Concentration**: Misclassifications cluster around overlapping or occluded gestures (`reaching behind`, `hair/makeup`, `talking left`), with confusion strongest in bbox-only variants.
- **FP–FN Trade-off**: False positives dominate between visually similar classes (e.g., `drinking` vs. `reaching behind`), while false negatives rise for rare actions when background context is removed.
- **Background Influence**: Full background aids separation of symmetric classes but encourages shortcuts; segmentation trims noise yet suppresses subtle cues, with bboxseg emerging as the most balanced compromise.

---

**Confidence-Stratified Clustering.** To align saliency analysis with model reliability, we stratify predictions into four confidence-based groups using $k$-means on softmax scores. As shown in Figure 5, correct predictions concentrate in high-confidence bins ($\geq 0.8$), while errors are more dispersed across mid-confidence ranges (0.4–0.6), indicating boundary-level uncertainty. Variants suppressing background (SEG, BBOXSEG) exhibit clearer separation of correct and incorrect groups than background-rich inputs (FULL, BBOX), reinforcing the diagnostic value of clustering.

---

**Cluster Analysis Takeaways**

- High-confidence clusters ($\geq 0.8$) align with correct predictions; errors dominate mid-confidence bins (0.4–0.6).
- False negatives cluster in uncertain regions, reflecting weak boundaries in overlapping activity classes.
- Background-suppressed variants improve separation, suggesting reduced reliance on spurious context.

---

**Saliency Tokenization under Cross-Domain Shift (Grad-CAM).** To complement performance and clustering analyses, we analyze EfficientNet-B0 with Grad-CAM across three background variants: BBOX, SEG, and BBOXSEG. Results shows that class-specific mosaics for representative *difficult/ambiguous* classes where cross-domain transfer degrades (e.g., midline symmetry, hand/face occlusions, rare actions). We use per-class tiles to align with the confusion and clustering diagnoses. All saliency maps use $\phi_{\text{ReLU}}$-normalized Grad-CAM with precomputed foreground/back-

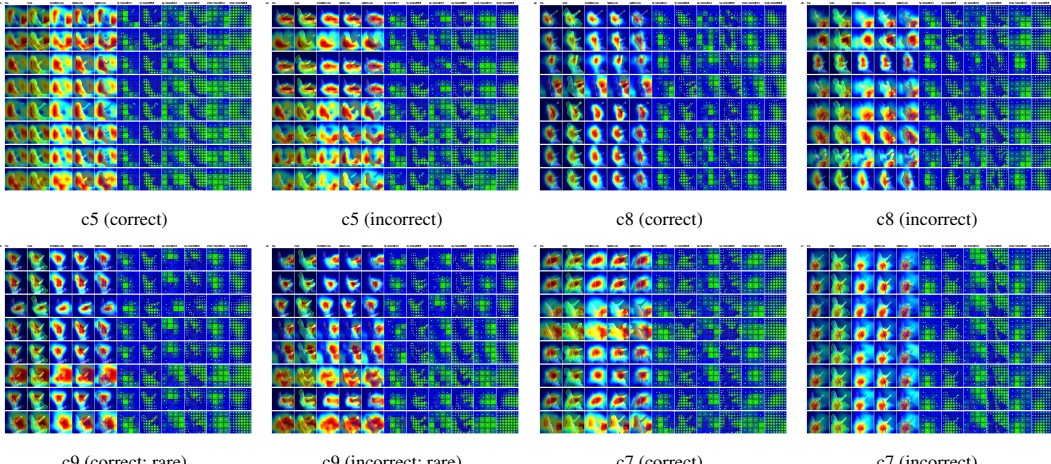

| c5 (correct) | c5 (incorrect) | c8 (correct) | c8 (incorrect) |

| c9 (correct; rare) | c9 (incorrect; rare) | c7 (correct) | c7 (incorrect) |

Figure 6: **SEG (Train SF3D-seg → Eval 100-Driver-seg)**. Removing background at both train and eval reduces shortcut saliency (fewer dashboard/seat activations) and tightens focus on hands/face. Remaining failures (right column) are dominated by occlusions and symmetric gestures consistent with confusion matrices (Fig. 3) and mid-confidence clusters.

ground masks and decomposed into foreground/background using YOLOv8 segmentation masks, then patchified into token grids ($4\times4$, $8\times8$) (Sec. 3).

**BBOX (SF3D→100-Driver).** When trained and tested with bounding-box crops, saliency reveals partial but noisy focus. Correct predictions (e.g., $c5, c7, c8$) localize on hands and torso, but error cases diffuse across dashboards and side panels, suggesting reliance on spurious context. This correlates with asymmetric confusion in Fig. 3 and the mid-confidence dispersion in clustering (Fig. 5). In short, BBOX does not fully suppress background, leaving shortcut attributions that undermine generalization.

**BBOXSEG (SF3D→100-Driver).** Joint use of bounding boxes with guided segmentation yields cleaner, more class-consistent saliency. Foreground activations are tighter and better aligned with causal regions (hands/face), while background leakage diminishes compared to BBOX. Rare actions ($c9$) still fragment under low data priors, but the overall reduction of spurious peaks supports BBOXSEG as a more balanced trade-off. These mosaics confirm the quantitative gains in Table 1, where BBOXSEG achieves best or near-best metrics across most settings.

**SEG (SF3D→100-Driver).** Foreground-only segmentation produces the sharpest saliency focus, with hands and faces dominating the maps. Correct predictions cluster strongly in high confidence ($\geq 0.8$), but errors for occluded or symmetric activities ($c7, c8, c9$) reveal fragmented saliency and missed cues. Removing all background reduces shortcut reliance but also trims subtle contextual signals that could disambiguate gestures, explaining both the improvement in robustness and the persistent rare-class collapse. This shows why SEG variants perform best in recall but sometimes lag in precision.

> **Saliency Class-Conditioned Insights (EffNet-B0, Grad-CAM)**
>
> **(i) Causal vs. spurious focus:** Correct predictions concentrate foreground saliency on driver hands/torso; errors exhibit diffuse background peaks (dashboards, seats), mirroring mid-confidence clusters (Fig. 5). **(ii) Background control:** BBOXSEG and SEG attenuate shortcut saliency relative to BBOX, improving class separation but not eliminating failures driven by occlusion/symmetry. **(iii) Error taxonomy:** (a) *Symmetric collapse* ($c7/c8$): left/right talk or similar gestures flip; maps straddle midline. (b) *Occlusion-driven misses* ($c5/c9$): hands/objects hidden; saliency fragments across torso/seat. (c) *Rare-class drift* ($c9$): weak priors amplify spurious background. **(iv) Tokenization payoff:** The observed fg/bg separation patterns motivate fg- and bg-token vectors; embedding these (Sec. 3) into VLM/vLLM spaces enables divergence diagnostics beyond accuracy.

**Tokenization analysis.** The panel in Fig. 7 illustrates how saliency maps are converted into structured descriptors. The $4\times4$ grid captures coarse localization, showing that entire arm/torso patches dominate in fg tokens with max values near 1.0. The $8\times8$ grid provides finer granularity: contiguous patches around the hand cluster show high mean ($\geq 0.7$) and sharp maxima, while background patches are scattered and low-intensity.

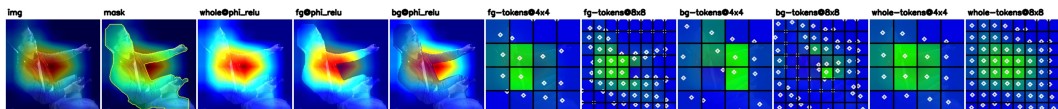

Figure 7: **Tokenization example** ($c0$, correct). Grad-CAM map normalized with $\phi_{\mathrm{ReLU}}$, decomposed into foreground (fg) and background (bg) using YOLOv8 masks, then patchified into $4{\times}4$ and $8{\times}8$ grids. Tokens encode per-patch mass, mean, max, centroid, and fg/bg fraction. Diamonds mark centroids; greener cells carry higher saliency mass.

Foreground vs. background comparison confirms that fg tokens are compact and class-discriminative, whereas bg tokens are noisy but occasionally spike, explaining residual shortcuts. In downstream use, fg tokens provide causal evidence for activity classification, bg tokens flag potential spurious cues, and whole-map tokens preserve global consistency. Embedding these token vectors into VLM/vLLM spaces (Sec. 3) enables divergence analysis: stable fg token embeddings indicate robust generalization, while unstable bg embeddings reveal shortcut reliance.

**Per-Class Saliency (Class $c7$: talking-left).** We compare class $c7$ saliency across BBOX, BBOXSEG, and SEG variants using the same exemplar image (img_116). All panels were normalized with $\phi_{\mathrm{ReLU}}$ Grad-CAM and decomposed into foreground/background with YOLOv8 masks. Token statistics ($4{\times}4$, $8{\times}8$) highlight distinct attribution profiles.

**BBOX (SF3D $\rightarrow$ 100-Driver).** Foreground saliency is high (mean $\approx 0.27$, max $\approx 1.0$), but background tokens retain non-negligible activation (mean $\approx 0.11$), particularly around seat/dash regions:contentReference[oaicite:0]index=0. This explains why confusion matrices show midline collapse between left/right talking classes: saliency COM drifts towards background tokens, reinforcing shortcut reliance.

**BBOXSEG (SF3D $\rightarrow$ 100-Driver).** Here, background suppression tightens the saliency map: foreground tokens dominate with mean $\approx 0.27$, background drops further ($\approx 0.11$):contentReference[oaicite:1]index=1. $4{\times}4$ grid reveals compact central patches (idx 5,9,10) carrying ¿0.6 mean saliency, aligning with driver torso/face. This supports improved separation seen in clustering (mid-confidence bins shrink).

**SEG (SF3D $\rightarrow$ 100-Driver).** Segmentation removes background almost entirely (fg mean $\approx 0.17$, bg mean $\approx 0.15$), yet also trims subtle context:contentReference[oaicite:2]index=2. $8{\times}8$ tokens reveal sparse activations with lower max values ($\approx 0.35$), fragmenting across patches instead of forming a single COM. This mirrors performance drops for occluded/symmetric gestures: model sees less spurious context, but also less discriminative signal.

**Cross-variant tokenization.** $4{\times}4$ tokens provide coarse localization (robust cluster-level analysis), while $8{\times}8$ tokens capture finer but noisier saliency dispersion. Foreground tokens stabilize under BBOXSEG; SEG disperses them, weakening boundary confidence. These token embeddings (fg/bg, $4{\times}4$ vs. $8{\times}8$) will be used downstream as inputs to VLM/vLLM divergence measures.

> **Class $c7$ (Talking-Left): Saliency Insights**
>
> **(i) BBOX** shows strong fg focus but residual bg leakage (dash/seat) drives midline confusions; **(ii) BBOXSEG** yields sharper fg attribution with compact high-saliency patches reducing spurious bg reliance; **(iii) SEG** removes most bg but fg tokens fragment, with symmetric gestures still collapsing; **(iv) Tokenization** confirms $4{\times}4$ grids capture stable coarse trends while $8{\times}8$ adds granularity yet noise, supporting fg/bg divergence embeddings.

*Scope note.* For efficientnet_b0_seg_190925_021131 we have only error mosaics at present (index artifacts); we therefore restrict to qualitative trends and defer per-class token statistics to future work scope. This does not affect the core claims (fg focus in correct; bg diffusion in errors) consistently observed across other runs.

## 5 DISCUSSION AND FUTURE WORK

This work set out to move saliency from qualitative visualization toward standardized, embeddable diagnostics. By decomposing Grad-CAM maps into foreground and background masses and further tokenizing them into patch descriptors, we provide a structured signal of how models attend to causal versus spurious regions. The analysis shows three recurring themes. **(i) Performance degradation is asymmetric across classes**: under domain shift, symmetric gestures and occlusion-heavy actions collapse disproportionately, while visually distinct activities transfer more reliably. **(ii) Saliency exposes diagnostic evidence beyond accuracy**: correct predictions concentrate on hands and torso, errors diffuse into dashboards and seats, and mid-confidence clusters reveal elevated background

mass. **(iii) Background control helps but is insufficient**: segmentation and bboxseg variants reduce shortcuts and stabilize token distributions, yet failures tied to symmetry, occlusion, and rare-class drift persist. Foreground-background tokenization elevates these observations into a reproducible representation: errors can be localized to token instabilities, and divergence between models can be linked to causal versus spurious focus. The framework therefore complements accuracy with a diagnostic dimension directly aligned with generalization. Although our case study centered on driver distraction detection, the implications extend to activity recognition broadly, where foreground cues often compete with contextual correlates in shaping transfer performance.

Crucially, these findings reflect back on our central hypothesis: that behavioral divergence between models is proportional to divergence in their saliency embeddings. The evidence supports this direction only partially. Foreground-background tokenization reveals consistent links between saliency divergence and model reliability, yet residual errors from symmetry and occlusion indicate that saliency embeddings alone cannot capture all behavioral variance. This critical gap reframes the hypothesis not as a closed claim but as a working diagnostic lens-one that must be stress-tested with embeddings in multimodal spaces and across broader domains.

**Future Work.** Several directions remain. (i) Embedding saliency tokens into multimodal encoders (VLMs, vLLMs) to quantify model divergence in semantic space. (ii) Extending tokenization to architectures with native patch structures (e.g., ViTs, hybrids). (iii) Leveraging token statistics for training interventions such as saliency-guided augmentation or background dropout. (iv) Improving fg/bg masks with stronger detectors or foundation models to reduce annotation bias. (v) Broadening benchmarks beyond SF3D and 100-Driver to establish generality across activity recognition tasks.

## 6 CONCLUSION

We introduced *Saliency Attribution for Goal-grounded Evaluation*, a framework that converts raw saliency into reproducible, tokenized diagnostics. By decomposing Grad-CAM maps into foreground and background masses and encoding them as patch-level descriptors, the framework exposes how domain shift drives asymmetric class collapse, mid-confidence background reliance, and failures from symmetry or occlusion. The results support our central hypothesis that divergence in saliency patterns tracks behavioral divergence between models, while also revealing limits where saliency embeddings alone cannot resolve ambiguity. This establishes a principled lens that complements accuracy with interpretable diagnostics and opens the path toward embedding saliency tokens in multimodal spaces to quantify and mitigate divergence in activity recognition.

## LARGE LANGUAGE MODELS USAGE

This work used large language models (LLMs), including ChatGPT (OpenAI), SciSpace, and Gemini, for writing polish, literature retrieval, initial research feasibility, research ideation, coding assistance (formatting and reviews), and LaTeX editing. All technical contributions and analyses were performed by the authors.

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

Table 2: Mapping of original 100-Driver labels to the unified ten-class schema and corresponding SF3D labels.

| Unified class | Original 100-Driver label(s) | SF3D label | Human-readable description |
|---|---|---|---|
| c0 | C1_Drive_Safe | c0,safe_driving | Normal Driving (C1) |
| c1 | C7_Text_Right | c1,texting_right | Texting Right (C7) |
| c2 | C5_Talk_Right | c2,talking_phone_right | Phone Right (C5) |
| c3 | C6_Text_Left | c3,texting_left | Texting Left (C6) |
| c4 | C4_Talk_Left | c4,talking_phone_left | Phone Left (C4) |
| c5 | C18_Operate_Radio | c5,operating_radio | Operate Radio (C18) |
| c6 | C16_Eat_Left | c6,drinking | Drink (C16) |
| c7 | C20_Reach_Behind | c7,reaching_behind | Reach Behind (C20) |
| c8 | C8_Make_Up | c8,hair_and_makeup | Make Up (C8) |
| c9 | C22_Talk_to_Passenger | c9,talking_to_passenger | Talk Passenger (C22) |

Kai Xiao, Logan Engstrom, Andrew Ilyas, and Aleksander Madry. Noise or Signal: The Role of Image Backgrounds in Object Recognition, June 2020.

Frank Zandamela, Terence Ratshidaho, Fred Nicolls, and Gene Stoltz. Cross-dataset performance evaluation of deep learning distracted driver detection algorithms. *MATEC Web Conf.*, 370:07002, 2022. ISSN 2261-236X. doi: 10.1051/matecconf/202237007002.

Ziwei Zheng, Junyao Zhao, Le Yang, Lijun He, and Fan Li. Spot Risks Before Speaking! Unraveling Safety Attention Heads in Large Vision-Language Models, January 2025.

## A  DATASET PREPARATION DETAILS

The State Farm Distracted Driver dataset (SF3D) (Montoya et al., 2016) and the 100-Driver dataset (Wang et al., 2023) are unified into a ten-class schema through a declarative JSON mapping. Forward mappings associate original labels (e.g., C7_Text_Right) with unified codes (e.g., c1), while reverse mappings preserve provenance, and human-readable labels provide semantic clarity. This mapping generalizes to any pair of datasets with partial overlap. An excerpt is shown below:

SEQUENTIAL CREATION PROCESS

Both datasets undergo the same reproducible workflow:

1. **Label mapping:** remap dataset labels into the unified schema using declarative JSON files. For 100-Driver, this produces train/val/test splits, remapped indices, mosaics, and per-class distributions.
2. **Variant generation:** run YOLOv8 segmentation to create `bbox`, `seg`, and `bboxseg` forms, each emitting cropped RGBs, masks, annotations, and logs under `logs/annotate-<timestamp>`.
3. **Index regeneration:** use `regen_loadertxt.sh` to rebuild train/val/test indices, ensuring splits remain consistent after filtering.
4. **Configuration update:** extend `ddd-datasets.yml` with dataset-IDs for each variant, pointing to regenerated indices and directories.
5. **Symlink creation:** map timestamped directories to canonical variant names (e.g., `sf3d-day-bydriver-bbox`) for stability.
6. **Summaries:** verify mosaics, per-class plots, and `summary.json` files capturing counts, fg/bg ratios, and error statistics.

DIRECTORY STRUCTURES

Figure 8 shows an abridged directory layout rendered using the `dirtree` package, closely mirroring the filesystem output of the `tree` command. SF3D uses timestamped annotation runs with symlinks to canonical names, while 100-Driver is organized by day/night and camera.

100-DRIVER-SF3D-NC10 HIERARCHY

After remapping the original 22-class 100-Driver dataset into the ten-class SF3D schema, the new dataset follows a reproducible directory layout. Each run is timestamped under `logs/annotate-<ddmmyy_hhmmss>` and contains annotation outputs, variants, metadata, and split indices. Figure 9 shows the abridged hierarchy.

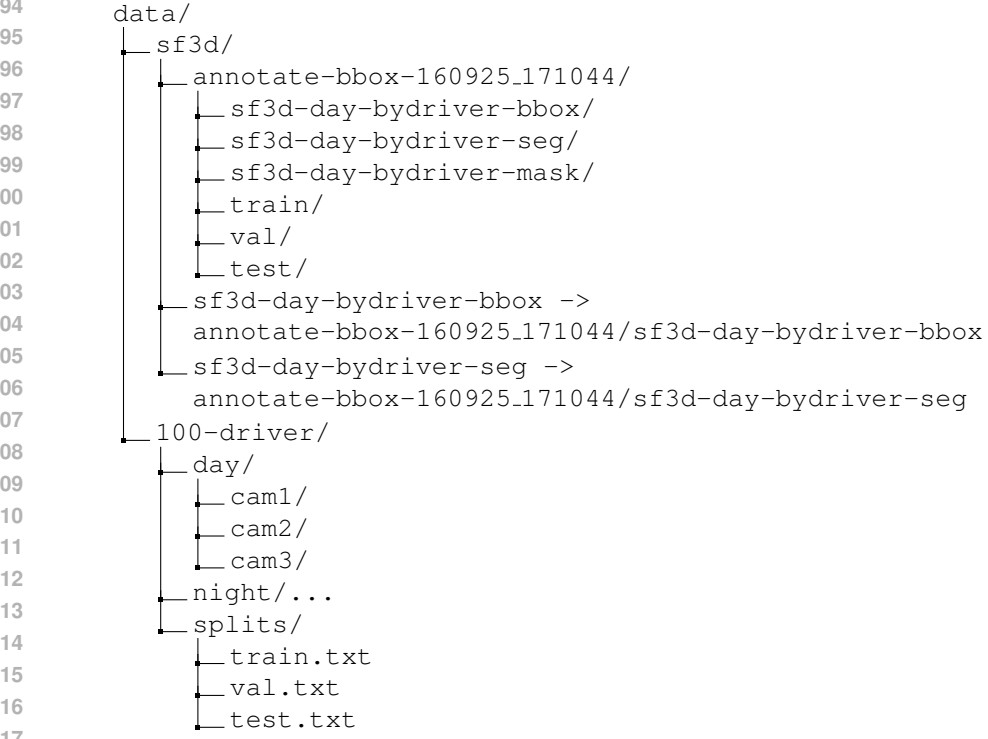

```
594    data/
595    |__sf3d/
596    |  |__annotate-bbox-160925_171044/
597    |  |  |__sf3d-day-bydriver-bbox/
598    |  |  |__sf3d-day-bydriver-seg/
599    |  |  |__sf3d-day-bydriver-mask/
600    |  |  |__train/
601    |  |  |__val/
602    |  |  |__test/
603    |  |__sf3d-day-bydriver-bbox ->
604    |  |  annotate-bbox-160925_171044/sf3d-day-bydriver-bbox
605    |  |__sf3d-day-bydriver-seg ->
606    |     annotate-bbox-160925_171044/sf3d-day-bydriver-seg
607    |__100-driver/
608       |__day/
609       |  |__cam1/
610       |  |__cam2/
611       |  |__cam3/
612       |__night/...
613       |__splits/
614          |__train.txt
615          |__val.txt
616          |__test.txt
617
```

Figure 8: Abridged directory layout for SF3D and 100-driver-sf3d-nc10. The straight-line style mirrors Unix `tree` output, left-aligned for readability.

The creation workflow relies on a small set of shell scripts. For SF3D and 100-Driver variants the following commands are executed sequentially:

1. **Annotation and segmentation:**
   ```
   bash scripts/anonflow.sf3d-day-bydriver.yolov8s-seg.sh
   bash scripts/anonflow.sf3d-day-bydriver-bbox.yolov8s-seg.sh
   ```

   These generate bbox, seg, and bboxseg variants under timestamped `logs/annotate-*`.

2. **Symlink updates:**
   ```
   vi scripts/datasets/sf3d.symlinks.sh
   ```

   Extend symlinks so canonical dataset-IDs point to the latest timestamped run.

3. **Split regeneration:**
   ```
   bash scripts/regen_loadertxt.sh
   ```

   Rebuild train/val/test indices to exclude filtered images and update the dataset configuration.

4. **Dataset configuration:** Edit `data/ddd-datasets.yml` to add or update entries for new dataset variants, pointing to regenerated indices.

ORIGINAL 100-DRIVER LABELS

For reference, the original label space of the 100-Driver dataset is illustrated in Fig. 10. This mosaic shows all native activity classes before remapping to the ten-class schema. In the main text , we cross-reference this appendix to highlight the semantic reduction.

STORAGE FOOTPRINTS

The processed SF3D variants occupy about 14 GB, while 100-driver-sf3d-nc10 variants occupy about 146 GB (90 GB day, 56 GB night). Each run directory includes lightweight `summary.json` files, mosaics, and distribution plots to ensure reproducibility without duplicating raw data.

```
logs/annotate-<ddmmyy_hhmmss>/
|__100-driver-day-cam2-annotation/
|__100-driver-day-cam2-bbox/
|__100-driver-day-cam2-mask/
|__100-driver-day-cam2-seg/
|__100-driver-day-cam2-viz/
|__modelinfo.json
|__summary.json
|__train/
|   |__imgs_list.csv
|   |__labels.csv
|   |__summary.json
|   |__summary.missed.json
|   |__train.csv
|__val/
|   |__imgs_list.csv
|   |__labels.csv
|   |__summary.json
|   |__summary.missed.json
|   |__val.csv
|__test/
    |__imgs_list.csv
    |__labels.csv
    |__summary.json
    |__summary.missed.json
    |__test.csv
```

Figure 9: Custom dataset hierarchy for 100-driver-sf3d-nc10 (Cam2 example). Each run contains annotation outputs, variants (bbox, seg, mask, viz), metadata files, and regenerated indices with CSV summaries.

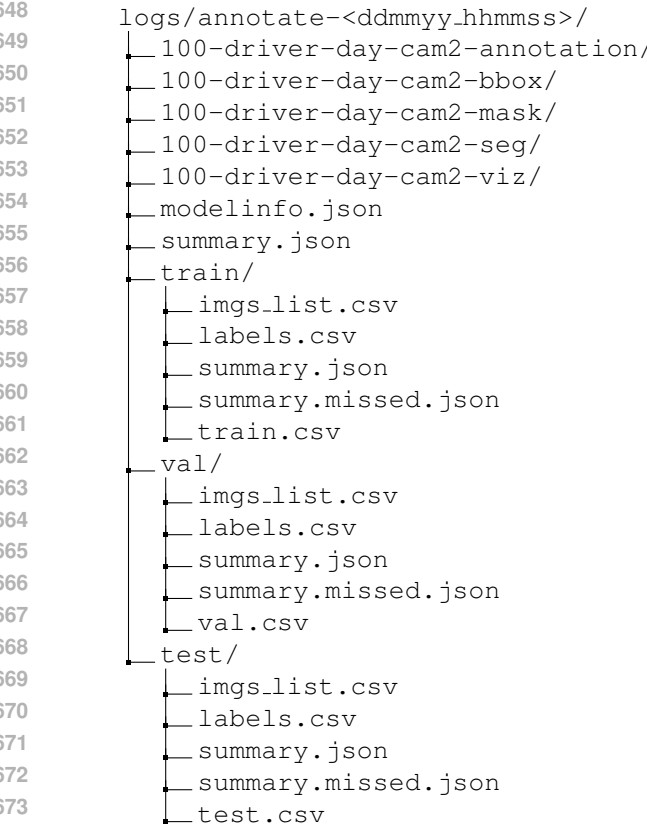
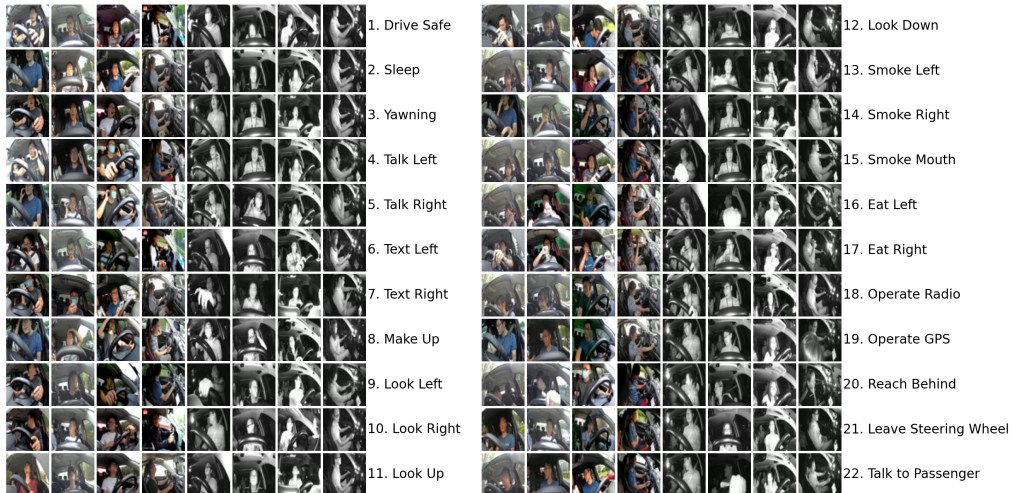

Figure 10: Original 100-Driver label space with day/night conditions and activity diversity, prior to remapping into the ten-class schema.

REPRODUCIBILITY CONVENTIONS

All datasets and variants are declared in ddd-datasets.yml, indices are regenerated post-filtering, and creation scripts emit standardized outputs under timestamped directories. Each run records mosaics, per-class counts, and foreground-background ratios for auditing.

## B  TRAINING SCRIPTS AND SCHEDULING

Training on each dataset variant is automated through reproducible shell scripts. These scripts loop over the chosen architectures (DenseNet-121, EfficientNet-B0, ResNet-18, ResNet-50, SqueezeNet-1.0), set common hyperparameters (ImageNet pretraining, 100 epochs, batch size 32, 10 output classes, cross-entropy loss), and log outputs in timestamped directories under `logs/`. All scripts invoke the same Python entry point (`python -m src.train`) with variant-specific dataset arguments.

### SF3D (BY-DRIVER PROTOCOL)

- `train.loop.m174.sf3d-day-bydriver-pretrain.sh` - baseline training on the FULL dataset.
- `train.loop.m174.sf3d-day-bydriver-bbox-pretrain.sh` - training on cropped bounding-box variants.
- `train.loop.m174.sf3d-day-bydriver-seg-pretrain.sh` - training on segmented foreground masks.
- `train.loop.m174.sf3d-day-bydriver-mask-pretrain.sh` - training on binary mask representations.
- `train.loop.m174.sf3d-day-bydriver-bboxseg-pretrain.sh` - training on bbox-guided segmentation variants.
- `run.train.loop.m174.sf3d-day-bydriver-pretrain.sh` - batch scheduler that sequentially executes the above scripts, enabling all variants to be trained in one call.

Each script defines the dataset variant by setting `dataset=$VARIANT` (e.g., `sf3d-day-bydriver-bbox`) and iterates over the architecture list. For each model, it constructs a unique output directory path of the form:

```
logs/train-CrossEntropyLoss_10-<timestamp>/
    <arch>-bs<batch_size>-nc10-<dataset>-<timestamp>/
```

where checkpoints, logs, and metrics are stored. Log files are simultaneously streamed to disk using `tee`, ensuring both console and persistent recording.

### 100-DRIVER-SF3D-NC10 (DAY/CAM2)

An equivalent set of scripts exists for the 100-driver-sf3d-nc10 dataset. The naming convention mirrors the SF3D case, with `100-driver-day-cam2-sf3d-nc10` replacing the dataset identifier. These scripts support the same five architectures, hyperparameters, and logging conventions, and include a batch runner to schedule all variants.

### USAGE NOTES

Scripts can be executed individually when fine-tuning a specific dataset–variant pair, or invoked through the batch runner to cover all variants in sequence. Epochs and batch size can be overridden at runtime by passing arguments:

```
bash scripts/train.loop.m174.sf3d-day-bydriver-pretrain.sh 50 64
```

which would train for 50 epochs with batch size 64. Each run is reproducible and traceable through the timestamped log directories.

### EXTENSIBILITY

While the main experiments report results for five representative CNNs, the training loop is designed to be fully generic. By editing the model list in the shell scripts, any of the following additional 21 torchvision models can be trained out-of-the-box under the same configuration:

Because the scripts pass the model name as an argument to the common training entry point, these architectures integrate seamlessly with the same preprocessing, logging, and evaluation pipeline. This makes the training framework both flexible and forward-compatible with new backbones added to torchvision.

Table 3: Additional torchvision models supported out-of-the-box by the training loop.

| Family | Models (input size) | |
|---|---|---|
| DenseNet | 161, 169, 201 | $(224 \times 224)$ |
| Inception | v3 | $(299 \times 299)$ |
| MobileNet | v2, v3-small, v3-large | $(224 \times 224)$ |
| ResNet | 34, 101, 152 | $(224 \times 224)$ |
| Wide-ResNet | 50_2, 101_2 | $(224 \times 224)$ |
| ShuffleNet | v2_x1_0 | $(224 \times 224)$ |
| VGG | 11, 11_bn, 13, 13_bn, 16, 16_bn, 19, 19_bn | $(224 \times 224)$ |

