# OpenReview forum: "SAGE Can Quantify Why Two Models Behave Differently"
_ICLR.cc/2026/Conference — ICLR 2026 Conference Withdrawn Submission_

### Official Review · Reviewer_LnNh · 2025-10-31

**Soundness:** 2
**Presentation:** 1
**Contribution:** 1
**Rating:** 2
**Confidence:** 3

**Summary:**

Generalization in activity recognition is difficult due to domain shift between controlled training data and in-the-wild data. It's not clear whether in-the-wild predictions depend on foreground cues which are "causal" in nature or on spurious background signals. The proposed works generalize the "Class Activation Mapping" frameworks which provided saliency wrt input (i.e. attribution maps) into a diagnostic framework.

**Strengths:**

Paper tackles an important problem, namely, assigning attribution for a black box prediction to the image regions, i.e. are the predictions being made causally tied to the image regions that are necessary to make such a prediction. As the authors point, class activation maps are attractive however can be misleading in nature. The authors propose to standardize the use of these activation maps to capture patch level statistics that can be correlated with downstream metrics like accuracy, which might be a plausible way of establishing some notion of causality and checking if spurious correlations exist.

**Weaknesses:**

The paper reads like a technical report and does not clearly outline their algorithm or method -- or the specific strengths of why we need their framework to diagnose mis-classifications in activity recognition. I advise the authors the following concrete suggestions, however, there are likely many more such changes that are needed before acceptance at ICLR --

(a) Demarcate and show why foreground object saliency is "causal" in nature and why is their tokenization framework necessary for "diagnostics". In general, the inputs, outputs and the end goal of their problem is not clearly defined in the paper and is hard to grasp even as a reader who is already well verse with the driver activity recognition task. For e.g. a person sitting behind the driver can trigger "talk_to_passenger" activity, thus if the saliency is not focussed on the driver (foreground in this case), that would make sense even if the saliency is focussed on the background.

(b) The figures are very difficult to read and are dumped on the reader to interpret in context of the work, they need to be made more accessible and tied to the proposed method. Some of the figures are also incomplete (some confusion matrices show the class name, versus others show the index). For e.g., in fig 5, while I would like to believe there is significance in understanding their confidence-stratified clusters, however, the paper doesn't motivate what the reader should be looking at and why they should be looking at these clusters in these three different forms (scatter, band and histogram plots) in the context of their method.

(c) I'm not sure what the specific claims made by the work are that was missing from prior work, and thus it is impossible to argue for or against the work. No comparisons are made to existing work or to any existing benchmark that would help situate their analysis. While I'm not against analysis papers, it is important for papers to describe the method succinctly and explain the kind of analysis that can be conducted given their new method.

**Questions:**

Please provide a concrete response to the weaknesses and also provide a concrete list of contributions that the reviewers should employ to evaluate this work.

---

### Official Review · Reviewer_LQEA · 2025-10-31

**Soundness:** 2
**Presentation:** 1
**Contribution:** 1
**Rating:** 2
**Confidence:** 3

**Summary:**

The authors address the important problem of generalization for Vision-based activity recognition and propose a framework for it. The framework unifies heterogeneous datasets via category alignment and sample balancing. It then produces controlled foreground and background variants, computes saliency maps, and tokenizes them for comparison using embedding-based methods. Thus, the framework provides a diagnostic means to identify how models utilize causal versus spurious regions by disentangling foreground and background saliency. The authors provide empirical evidence to support their claims.

**Strengths:**

1) The authors address the important problem of cross domain generalization in Vision-based activity recognition.

2) The design of experiment is well-structured to isloate background influenceby using four variants (FULL, BBOX, SEG, and BBOXSEG).

3) Additionally the authors evaluates multiple CNN architectures, domains and integrates quantitative diagnostics in a unified pipeline.

**Weaknesses:**

1) Although the framework is well-engineered and focuses on reproducibility yet the core operations (like spatial normalization, foreground/background separation, etc) are well-established in prior works. Hence, the conceptual novelty appears limited, as the primary contribution seems to be building the system and conducting large-scale analysis, rather than proposing a new methodological or theoretical advancement.

2) The reported accuracies in the Results section are low. A within-domain baseline would help readers to better contextualize the reported numbers.

3) The numbers in the confusion matrix of Figure 3 are barely visible.

4) The paper is hard to follow, especially the Results section. While the authors have presented a large amount of experimental evidence but the results section is somewhat difficult to follow, as key findings and their implications become clear only in the Discussion section.

**Questions:**

I request the authors to address the identified weaknesses above. I would be open to revising my evaluation if these issues are effectively resolved.

---

### Official Review · Reviewer_NX3V · 2025-11-04

**Soundness:** 2
**Presentation:** 3
**Contribution:** 2
**Rating:** 4
**Confidence:** 4

**Summary:**

This paper introduces SAGE, a framework designed to move beyond qualitative saliency visualizations by converting them into tokenized representations. The core idea is that behavioral divergence between models under domain shift can be diagnosed by comparing their saliency embeddings. The authors demonstrate the framework on a driver distraction detection task, unifying two datasets (StateFarm and 100-Driver) into a 10-class schema and generating various foreground/background variants (FULL, BBOX, SEG, BBOXSEG). They benchmark several CNN models, analyze cross-domain generalization, and show that saliency tokenization can reveal whether a model relies on causal foreground cues or spurious background signals. The primary contribution is a modular pipeline for standardized saliency diagnostics.

**Strengths:**

The paper presents a well-structured and reproducible pipeline for saliency analysis, which is a valuable contribution to the explainability toolbox. The idea of tokenizing saliency maps into structured, patch-level descriptors is novel and aligns well with modern patch-based architectures like Vision Transformers, providing a clear path for future integration. The experimental design is rigorous, featuring a systematic cross-dataset and cross-variant evaluation that cleanly isolates the effect of background context on model generalization. The comprehensive analysis—spanning performance metrics, confidence-stratified clustering, and qualitative saliency mosaics—provides compelling evidence that foreground-background saliency disentanglement offers a diagnostic signal complementary to accuracy. The framework is modular and generalizable, promising applicability beyond the specific task studied here.

**Weaknesses:**

The most significant weakness is the disconnect between the paper's stated motivation and its actual execution. Despite prominently featuring "Vision-Language Models (VLMs)" in the abstract and keywords, and framing tokenization as being "aligned with tokenized vision architectures such as ViTs and VLMs," the work contains zero experiments or analysis involving a Vision-Language Model. The entire study is conducted using standard CNNs (e.g., EfficientNet-B0). This renders the core motivational premise unfulfilled and undermines the paper's relevance to the VLM community.

In addition, the methodological contribution is incremental. The process of generating saliency maps (using established tools like Grad-CAM), segmenting foreground/background, and computing patch statistics is a straightforward engineering integration of existing techniques rather than a novel algorithmic advancement. The resulting "token" is simply a vector of hand-crafted statistical features (mass, mean, max, centroid), lacking the semantic richness one would expect from a representation designed for VLMs.

The paper reads more like a well-documented technical report or a system description. It extensively details the data preparation and experimental pipeline but offers limited new theoretical insight. The central hypothesis—that model behavioral divergence is proportional to saliency embedding divergence—is only partially supported by the presented analysis of CNNs. The failure to demonstrate this with the very architectures (VLMs) that would make the tokenization most powerful is a critical omission.

**Questions:**

Please refer to my weaknesses, especially the first concern regarding vlm validation.

---

### Official Review · Reviewer_3aJK · 2025-11-16

**Soundness:** 2
**Presentation:** 2
**Contribution:** 1
**Rating:** 2
**Confidence:** 4

**Summary:**

The paper introduces a framework for saliency-based driver distraction recognition. The key idea is to distinguish between foreground and background in order to establish how a saliency model determines signals to attend to over spurious regions. The paper further proposes creating a 10-class variant of the Statefarm and 100-Driver datasets to highlight the challenges of identifying salient action over background control. Some early experiments show cross-domain generation between models trained on classes from one data and evaluated on another.

**Strengths:**

Overall, the paper discusses the important problem of driver distraction recognition, which can be considered as an activity recognition task across multiple classes of distraction. Furthermore, the experimental setup is quite rigorous, evaluating a large variety of model backbones. Lastly, the cross-domain evaluation is quite interesting, showing the ability of a model to generalize across multiple datasets for detecting driver attention/distraction.

**Weaknesses:**

There are several issues with the paper in its current state:
- It is unclear how saliency is a strong reflection of driver's activity. Specially when distinguishing between background and foreground, it seems like the driver will always occupy the foreground, regardless of what the type of distraction is (or is not). Explanation regarding this choice over others (say driver pose recognition) is missing in the paper.
- In addition to above, the paper lacks several details and feels incomplete. While the motivation of the approach is sound, there are not enough diverse experiments to understand whether the proposed multi-background saliency variants (FULL vs BBOX vs SEG vs BBOXSEG) are sufficient.
- The paper mentions using ViT and other VLM models being used, but there is no evaluation conducted to support this claim in either the table or the figures.
- The writing of the paper could improve significantly. Details regarding CNN backbones can go to the appendix, allowing for more space to discuss detailed experiments and evaluations. Additionally, the framework section does not need to introduce so many variables, making it hard to follow.
- Minor: There are several typos in the paper like in L139 and L184.

**Questions:**

The reviewer requests the authors to read the things mentioned in the Weaknesses section, and possibly address them in a later version of the paper with more thorough analysis and experimental results.

---

### Note · Authors · 2025-12-02

**Comment:**

We thank the reviewers and the area chair for their time and feedback. After internal discussion, we have decided to withdraw the submission.

**Withdrawal Confirmation:**

I have read and agree with the venue's withdrawal policy on behalf of myself and my co-authors.